# AN INFORMATION THEORETIC APPROACH TO INTERACTION GROUNDED LEARNING

## ABSTRACT

Reinforcement learning (RL) problems where the learner attempts to infer an unobserved reward from some feedback variables have been studied in several recent papers. The setting of *Interaction-Grounded Learning (IGL)* is an example of such feedback-based reinforcement learning tasks where the learner optimizes the return by inferring latent binary rewards from the interaction with the environment. In the IGL setting, a relevant assumption used in the RL literature is that the feedback variable $Y$ is conditionally independent of the context-action $(X, A)$ given the latent reward $R$. In this work, we propose *Variational Information-based IGL (VI-IGL)* as an information-theoretic method to enforce the conditional independence assumption in the IGL-based RL problem. The VI-IGL framework learns a reward decoder using an information-based objective based on the conditional mutual information (MI) between the context-action $(X, A)$ and the feedback variable $Y$ observed from the environment. To estimate and optimize the information-based terms for the continuous random variables in the RL problem, VI-IGL leverages the variational representation of mutual information and results in a min-max optimization problem. Furthermore, we extend the VI-IGL framework to general $f$-Information measures in the information theory literature, leading to the generalized $f$-VI-IGL framework to address the RL problem under the IGL condition. Finally, we provide the empirical results of applying the VI-IGL method to several reinforcement learning settings, which indicate an improved performance in comparison to the previous IGL-based RL algorithm.

## 1 INTRODUCTION

In several applications of reinforcement learning (RL) algorithms, the involved agent lacks complete knowledge of the reward variable, e.g. in applications concerning brain-computer interface (BCI) (Schalk et al., 2004; Serrhini & Dargham, 2017) and recommender systems (Maghakian et al., 2023). In such RL settings, the lack of an explicit reward could lead to a challenging learning task where the learner needs to infer the unseen reward from observed feedback variables. The additional inference task for the reward variable could significantly raise the computational and statistical complexity of the RL problem. Due to the great importance of addressing such RL problems with a misspecified reward variable, they have been exclusively studied in several recent papers (Xie et al., 2021; 2022; Maghakian et al., 2023).

To handle the challenges posed by a misspecified reward variable, Xie et al. (2021; 2022) propose the *Interaction-Grounded Learning (IGL) framework*. According to the IGL framework, the agent observes a multidimensional *context vector* based on which she takes an *action*. Then, the environment generates a *latent 0-1 reward* and reveals a multidimensional *feedback vector* to the agent. The agent aims to maximize the (unobserved) return by inferring rewards from the interaction, a sub-task which needs to be solved based on the assumptions on the relationship between reward and feedback variables.

As a result, the key to addressing the IGL-based RL problem is a properly inferred *reward decoder* $\psi \in \Psi$, which maps a context-action-feedback tuple $(X, A, Y) \in \mathcal{X} \times \mathcal{A} \times \mathcal{Y}$ to a prediction of the posterior probability on the latent reward $R$. Given such a reward decoder, the optimal policy can be obtained using standard contextual bandit algorithms (Langford & Zhang, 2007; Dudík et al., 2014). However, such a reward decoder will be information-theoretically infeasible to learn without addi-

tional assumptions (Xie et al., 2022). Consequently, the existing works on the IGL setting (Xie et al., 2021; 2022) make relevant assumptions on the statistical relationship between the random variables of context $X$, action $A$, feedback $Y$, and latent reward $R$. In particular, a sensible assumption on the connection between $X, A, Y, R$ is the following conditional independence assumption proposed by Xie et al. (2021):

**Assumption 1** (Full conditional independence). *For arbitrary $(X, A, R, Y)$ tuple where $R$ and $Y$ are generated based on the context-action pair $(X, A)$, the feedback $Y$ is conditionally independent of $X$ and $A$ given the latent reward $R$, i.e., $Y \perp\!\!\!\perp X, A | R$.*

In the work of Xie et al. (2021), a reward decoder $\psi : \mathcal{Y} \mapsto [0, 1]$ takes the feedback $Y \in \mathcal{Y}$ as input and outputs a prediction of the posterior distribution $\mathbb{P}(R = 1 | Y)$. Their proposed approach performs a joint training of the policy and the decoder by maximizing the difference in the decoded return between the learned policy and a "bad" policy that is known to have a low (true) return. Under Assumption 1, they show that the optimal policy with the maximum return can be learned statistically efficiently. However, the observation of the feedback variable is often under significant noise levels in practice, e.g. in the BCI application. In such noisy settings, Assumption 1 may still hold under an independent noise from the discussed random variables or may not hold when the noise is correlated with the context or action variables. Consequently, the discussed IGL-based methods may no longer achieve optimal results under such noisy feedback conditions.

In this paper, we attempt to address the mentioned challenges in the IGL-based RL problem and propose *Variational Information-based IGL (VI-IGL)* as an information-theoretic approach to IGL-based RL tasks. The proposed VI-IGL methodology is based on the properties of information measures that allow measuring the dependence among random variables. According to these properties, Assumption 1 will hold, i.e., the feedback variable $Y$ is conditionally independent of the context-action $(X, A)$ given the latent reward $R$, *if and only if* the conditional mutual information (CMI) $I(Y; X, A | R)$ is zero. Therefore, we suggest an information bottleneck-based approach (Tishby et al., 2000) and propose to learn a reward decoder via the following information-based objective value where $\beta > 0$ is a tuning parameter and $R_\psi$ is the random decoded reward from $\psi$:

$$\arg\min_{\psi \in \Psi}\{I(Y; X, A | R_\psi) - \beta \cdot I(X, A; R_\psi)\} \tag{1}$$

Intuitively, minimizing the first term $I(Y; X, A | R_\psi)$ ensures that the solved reward decoder satisfies the full conditional independence assumption. In addition, the second term $I(R_\psi; X, A)$ serves as a regularization term preventing the reward decoder from "over-fitting" to the feedback $Y$, and hence being more robust to the noise in the feedback variable.

Nevertheless, the objective function in (1) is challenging to optimize, since a first-order optimization of this objective requires estimating the value and derivatives of the MI for continuous random variables of the context $X$ and the feedback $Y$. To handle this challenge, we leverage the variational representation of MI (Donsker & Varadhan, 1983; Nguyen et al., 2010) and cast Objective (1) as a min-max optimization problem that gradient-based algorithms can efficiently solve. Using the variational formulation of the information-based objective, we propose the Variational Information-based IGL (VI-IGL) minimax learning algorithm for solving the IGL-based RL problem. The VI-IGL method applies the standard gradient descent ascent algorithm to optimize the min-max optimization problem following the variational formulation of the problem.

We numerically evaluate the proposed VI-IGL method on several RL tasks. Our empirical results suggest that VI-IGL can outperform the existing IGL RL algorithm under the presence of a noisy feedback variable. The main contributions of this paper can be summarized as:

1. We propose an information-theoretic approach to the IGL-based RL problem, which learns a reward decoder by minimizing an information-based objective function.

2. To handle the challenges in estimating and optimizing ($f$-)MI for continuous random variables, we leverage the variational representation and formulate our objective as a min-max optimization problem, which can be solved via gradient-based optimization methods.

3. We extend the proposed approach to $f$-Variational Information-based IGL ($f$-VI-IGL), leading to a family of algorithms to solve the IGL-based RL task.

4. We provide empirical results indicating that $f$-VI-IGL performs successfully compared to existing IGL-based RL algorithms.

## 2   RELATED WORKS

**Interaction-Grounded Learning (IGL).** The framework of IGL is proposed by Xie et al. (2021) to tackle learning scenarios without explicit reward. At each round, the agent observes a multidimensional context, takes an action, and then the environment generates a latent 0-1 reward and outputs a multidimensional feedback. The agent aims to optimize the expected return by observing only the context-action-feedback tuple during the interaction. When the feedback is independent of both the context and the action given the latent reward (full conditional independence), Xie et al. show that the optimal policy can be sample-efficiently learned with additional assumptions. To relax the full conditional independence requirement, Xie et al. (2022) introduce Action-Inclusive IGL, where the feedback can depend on both the latent reward and the action. They propose a contrastive learning objective and show that the latent reward can be decoded under a symmetry-breaking procedure. Recently, Maghakian et al. (2023) apply the IGL paradigm with a multi-state latent reward to online recommender systems. Their proposed algorithm is able to learn personalized rewards and show empirical success.

**Information-Theoretic Reinforcement Learning Algorithms.**   Reinforcement learning (RL) is a well-established framework for agents' decision-making in an unknown environment (Sutton & Barto, 2018). Several recent works focus on designing RL algorithms by exploiting the information-related structures in the learning setting. To perform exploration and sample-efficient learning, Russo and Van Roy (2014) propose information-directed sampling (IDS), where the agent takes actions that either with a small *regret* or yield large *information gain*, which is measured by the mutual information between the optimal action and the next observation. They show that IDS preserves numerous theoretical guarantees of Thompson sampling while offering strong performance in the face of more complex problems. In addition, information-theoretic approaches have been applied for *skills discovery* in machine learning contexts. Gregor, Rezende, and Wierstra (2016) introduce variational intrinsic control (VIC), which discovers useful and diverse behaviors (i.e., *options*) by maximizing the mutual information between the options and termination states. A setting that is close to our paper is using information-based methodology to learn reward functions in *inverse reinforcement learning* (IRL) (Ng & Russell, 2000). Levine, Popović, and Koltun (2011) propose to learn a cost function by maximizing the entropy between the corresponding optimal policy and human demonstrations. However, IGL is different from this setting, since it does not make any assumptions on the optimality of the observed behavior.

**Estimation of Mutual Information (MI).** Mutual information (MI) is a fundamental information-theoretic quantity that measures "the amount of information" between random variables. However, estimating MI in continuous settings is statistically and computationally challenging (Gao et al., 2015). Building upon the well-known characterization of the MI as the Kullback-Leibler (KL-) divergence (Kullback, 1997), recent works propose to use the variational representation of MI for its estimation and more generally for $f$- divergences (Nguyen et al., 2010; Belghazi et al., 2018; Molavipour et al., 2020).

## 3   PRELIMINARIES

### 3.1   INTERACTION-GROUNDED LEARNING (IGL)

In the Interaction-Grounded Learning (IGL) paradigm, at each round, a multidimensional *context* $x \in \mathcal{X}$ is drawn from a distribution $d_0$ and is revealed to the agent. Upon observing $x$, the agent takes action $a \in \mathcal{A}$ from a finite action space. Let $\Delta_{\mathcal{S}}$ denote the probability simplex on space $\mathcal{S}$. Given the context-action pair $(x, a)$, the environment generates a *latent and binary reward* $r \sim R(x, a) \in \Delta_{\{0,1\}}$ and returns a multidimensional *feedback* $y \in \mathcal{Y}$ to the agent. It can be seen that IGL recovers a contextual bandit (CB) problem (Langford & Zhang, 2007) if the reward is observed. Let $\pi \in \Pi : \mathcal{X} \mapsto \Delta_{\mathcal{A}}$ denote any stochastic policy. The expected return of policy $\pi$ is given by $V(\pi) := \mathbb{E}_{x \sim d_0} \mathbb{E}_{a \sim \pi(\cdot|x)}[\mu(x, a)]$, where $\mu(x, a)$ is the expected (latent) reward of any context-action pair $(x, a) \in \mathcal{X} \times \mathcal{A}$. The agent aims to learn the optimal policy, that is, $\pi^* := \arg\max_{\pi \in \Pi} V(\pi)$ while only observing the context-action-feedback tuple $(x, a, y)$ at each round of interaction.

## 3.2 $(f\text{-})$CONDITIONAL MUTUAL INFORMATION

The $(f\text{-})$mutual information (MI) (Ali & Silvey, 1966) is a standard measure of dependence between random variables in information theory. Formally, let $f : \mathbb{R}_+ \mapsto \mathbb{R}$ be a convex function satisfying $f(1) = 0$. The $f$-MI (Csiszár, 1967) between a pair of random variables $Z_1$ and $Z_2$ is given by

$$I_f(Z_1; Z_2) := D_f(\mathbb{P}_{Z_1 Z_2} \| \mathbb{P}_{Z_2} \otimes \mathbb{P}_{Z_1}). \tag{2}$$

In this definition, $D_f(\mathbb{P}\|\mathbb{Q})$ denotes the $f$-divergence between distributions $\mathbb{P}$ and $\mathbb{Q}$ defined as

$$D_f(\mathbb{P}\|\mathbb{Q}) := \mathbb{E}_{\mathbb{Q}} \left[ f\left( \frac{d\mathbb{P}}{d\mathbb{Q}} \right) \right]$$

Note that the standard KL-based conditional mutual information, which is denoted by $I(Z_1; Z_2)$, is given by $f(x) = x \log x$. Another popular $f$-divergence is Pearson-$\chi^2$ (Peason, 1900), where $f(x) = (x-1)^2$. An important property of $f$-MI is that two random variables $Z_1, Z_2$ are statistically independent if and only $I_f(Z_1; Z_2) = 0$, and hence dependence among between random variables can be measured via an $f$-mutual information.

Furthermore, the $f$-conditional MI (Csiszár, 1967) between a pair of random variables $Z_1$ and $Z_2$ when $Z_3$ is observed can be defined as

$$I_f(Z_1; Z_2 | Z_3) := D_f(\mathbb{P}_{Z_1 Z_2 | Z_3} \| \mathbb{P}_{Z_2 | Z_3} \otimes \mathbb{P}_{Z_1 | Z_3}). \tag{3}$$

Similarly, the standard KL-based conditional mutual information, denoted by $I(Z_1; Z_2 | Z_3)$, is given by $f(x) = x \log x$. One useful property of the $f$-CMI is that, if $Z_1$ is conditionally independent of $Z_2$ given $Z_3$ then it holds that $I_f(Z_1; Z_2 | Z_3) = 0$.

## 4 VARIATIONAL INFORMATION-BASED IGL

### 4.1 INFORMATION-BASED IGL

In this section, we derive an information-theoretic formulation for the IGL-based RL problem. As discussed earlier, in information theory, a standard measure of the (conditional) dependence between random variables is (conditional) mutual information (MI). Particularly, Assumption 1 (i.e., $Y \perp\!\!\!\perp X, A | R$) is equivalent to that the conditional MI between the context-action $(X, A)$ and the feedback variable $Y$ is zero given the latent reward $R$, i.e., $I(Y; X, A | R) = 0$.

Therefore, an information-theoretic approach is to learn a reward decoder $\psi \in \Psi : \mathcal{X} \times \mathcal{A} \times \mathcal{Y} \mapsto [0, 1]$ which minimizes the dependence measure $I(Y; X, A | R_\psi)$. Here, $R_\psi \sim$ Bernoulli$(\psi(X, A, Y))$ is the decoded 0-1 reward. On the other hand, note that the chain rule of MI results in the following identity

$$I(Y; X, A | R_\psi) = I(Y; X, A, R_\psi) - I(Y; R_\psi).$$

As a result of the above information-theoretic identity, training a reward decoder to minimize only $I(Y; X, A | R_\psi)$ will "over-fit" to the feedback $Y$ to maximize $I(Y; R_\psi)$, and hence may underperform under a noisy feedback variable. [Indeed, we can show that when minimizing only $I(Y; X, A | R_\psi)$: (A1) *any* feedback-dependent reward decoder $\psi : \mathcal{Y} \to [0, 1]$ attains a small value, and (A2) the minimum is attained by a set of deterministic feedback-dependent reward decoders $\psi : \mathcal{Y} \to \{0, 1\}$ for environments where the feedback variable $Y$ is (nearly) deterministic to the context-action $(X, A)$. Both cases can lead to the "over-fitting" problem. (The theoretical analysis can be found in Appendix B.)]

To address this overfitting issue, we propose the following regularized information-based IGL objective where $\beta > 0$ is a tunable parameter:

$$\arg\min_{\psi \in \Psi} \{ I(Y; X, A | R_\psi) - \beta \cdot I(X, A; R_\psi) \}. \tag{4}$$

In the optimization of the above objective function, minimizing the first term $I(Y; X, A | R_\psi)$ guides the reward decoder to satisfy the conditional independence assumption. Furthermore, the second term $I(X, A; R_\psi)$ will play the role of a regularization term biasing the reward decoder to carry higher information about context-action $(X, A)$, and hence preventing it from over-fitting to the

potentially-noisy feedback variable. To gain intuition on why Objective (4) can be robust against noisy feedback, we note that this problem formulation translates into the following problem

$$\arg\max_{\psi \in \Psi}\{I(X, A; R_\psi) - \beta^{-1} \cdot I(Y; X, A|R_\psi)\}$$

Thus, we can alternatively interpret Objective (4) as maximizing $I(X, A; R_\psi)$ under the full conditional independence constrain that $I(Y; X, A|R_\psi) = 0$, where $\beta^{-1}$ is the coefficient of the penalty term on $I(Y; X, A|R_\psi)$. Hence, under a proper selection of $\beta$, the noises present in context and action variables cannot significantly affect the accuracy of the optimized reward decoder. As demonstrated by our numerical results in Section 6.2, introducing this regularizer not only helps to handle a noisy feedback variable, but also results in a more consistent algorithm performance under lower noise levels.

## 4.2 VARIATIONAL INFORMATION-BASED IGL

While the previous sub-section introduces an information-theoretic objective to address the IGL-based RL problem, optimizing (4) in complex environments can be highly challenging. The primary challenge to solve (4) is that it requires estimating MI among continuous random variables of the context $X$ and the feedback $Y$, which is widely recognized as a statistically and computationally difficult problem (Paninski, 2003). To derive a tractable optimization problem, we utilize the variational representation of the KL-divergence, which reduces the evaluation and estimation of MI to an optimization task.

**Proposition 2** (Variational representation of KL-divergence Nguyen et al. (2010)). *Let $\mathbb{P}, \mathbb{Q} \in \Delta_\mathcal{S}$ be two probability distributions on space $\mathcal{S}$. Then,*

$$D_{\mathrm{KL}}(\mathbb{P}\|\mathbb{Q}) = \mathbb{E}_\mathbb{P}\left[\log\left(\frac{d\mathbb{P}}{d\mathbb{Q}}\right)\right] \geq \sup_{T \in \mathcal{T}}\left\{\mathbb{E}_{s\sim\mathbb{P}}[T(s)] - \mathbb{E}_{s\sim\mathbb{Q}}\left[e^{T(s)-1}\right]\right\}$$

Recall that the MI between random variables $Z_1 \in \mathcal{Z}_1$ and $Z_2 \in \mathcal{Z}_2$ is the KL-divergence between their joint distribution $\mathbb{P}_{Z_1 Z_2}$ and the product of their marginal distributions $\mathbb{P}_{Z_1} \otimes \mathbb{P}_{Z_2}$, i.e., $I(Z_1; Z_2) = D_{\mathrm{KL}}(\mathbb{P}_{Z_1 Z_2}\|\mathbb{P}_{Z_1} \otimes \mathbb{P}_{Z_2})$. Proposition 2 enables us to estimate $I(Z_1; Z_2)$ through optimizing over a class of function $T : \mathcal{Z}_1 \times \mathcal{Z}_2 \mapsto \mathbb{R}$. Therefore, we propose the *variational information-based IGL (VI-IGL)* optimization problem to solve Objective (4).

**Theorem 3** (VI-IGL optimization problem). *Objective (4) is equivalent to the following min-max optimization:*

$$\arg\min_{\psi \in \Psi} \max_{G \in \mathcal{G}} \min_{T \in \mathcal{T}}\left\{\mathbb{E}_{\mathbb{P}_{XAYR_\psi}}[G] - \mathbb{E}_{\mathbb{P}_{Y|R_\psi} \otimes \mathbb{P}_{XAR_\psi}}\left[e^{G-1}\right]\right.$$
$$\left. - \beta \cdot (\mathbb{E}_{\mathbb{P}_{XAR_\psi}}[T]) - \mathbb{E}_{\mathbb{P}_{XA} \otimes \mathbb{P}_{R_\psi}}\left[e^{T-1}\right])\right\} \tag{5}$$

*where $G \in \mathcal{G} : \mathcal{X} \times \mathcal{A} \times \mathcal{Y} \times \{0, 1\} \mapsto \mathbb{R}$ and $T \in \mathcal{T} : \mathcal{X} \times \mathcal{A} \times \{0, 1\} \mapsto \mathbb{R}$.*

The optimization problem (5) possesses three levels: [(i) the inner level minimizes over function class $T \in \mathcal{T}$ to estimate $I(X, A; R_\psi) = D_{\mathrm{KL}}(\mathbb{P}_{XAR_\psi}\|\mathbb{P}_{XA} \otimes \mathbb{P}_{R_\psi}) = \sup_{T \in \mathcal{T}}\{\mathbb{E}_{\mathbb{P}_{XAR_\psi}}[T(X, A, R_\psi)] - \mathbb{E}_{\mathbb{P}_{XA} \otimes \mathbb{P}_{R_\psi}}[\exp(T(X, A, R_\psi) - 1)]\}$ by Proposition 2, (ii) the medium level corresponds to maximizing over function class $G \in \mathcal{G}$ to estimate $I(Y; X, A|R_\psi) = D_{\mathrm{KL}}(\mathbb{P}_{XAYR_\psi}\|\mathbb{P}_{Y|R_\psi} \otimes \mathbb{P}_{XAR_\psi}) = \sup_{G \in \mathcal{G}}\{\mathbb{E}_{\mathbb{P}_{XAYR_\psi}}[G(X, A, Y, R_\psi)] - \mathbb{E}_{\mathbb{P}_{Y|R_\psi} \otimes \mathbb{P}_{XAR_\psi}}[\exp(G(X, A, Y, R_\psi) - 1)]\}$], and (iii) the outer level is a minimization problem to find the desired reward decoder. Specifically, the inner max-min optimization problem can be solved by simultaneously updating the variables over $\mathcal{G}$ and $\mathcal{T}$. Given the estimated values, the outer minimization problem searches for the optimal reward decoder. In practice, the functions $T$ and $G$ and the reward decoder $\psi$ can be parameterized using deep neural networks that can express complex functions. Through the application of expressive neural networks, we can efficiently parameterize and solve Objective (5) by alternatively updating the parameters using a gradient descent/ascent method.

## 5 THE $f$-VI-IGL ALGORITHM

### 5.1 THE EXTENDED $f$-VARIATIONAL INFORMATION-BASED IGL

In this section, we first propose an extended version of the information-based objective in (4) and the VI-IGL optimization problem (5). Recall that $f$-mutual information defined in Equation (3) generalizes the standard KL-divergence-based MI to a general $f$-divergence-based MI. Therefore, we can extend the standard MI-based IGL objective (4) to the following $f$-MI-based IGL objective:

$$\psi^* := \underset{\psi \in \Psi}{\arg\min} \{ I_{f_1}(Y; X, A | R_\psi) - \beta \cdot I_{f_2}(X, A; R_\psi) \} \tag{6}$$

where $f_1$ and $f_2$ are two $f$-divergences. Note that Objective (4) is a special case of the above formulation by selecting $f_1(x) = f_2(x) = x \log x$ to obtain the standard KL-based mutual information. Similar to the VI-IGL problem formulation, to derive a tractable optimization problem corresponding to the above task, we adopt the variational representation of $f$-divergences.

**Proposition 4** (Variational representation of $f$-divergences (Nguyen et al., 2010)). *Let $f : \mathbb{R}_+ \mapsto \mathbb{R}$ be a convex, lower-semicontinuous function satisfying $f(1) = 0$. Consider $\mathbb{P}, \mathbb{Q} \in \Delta_\mathcal{S}$ as two probability distributions on space $\mathcal{S}$. Then,*

$$D_f(\mathbb{P}\|\mathbb{Q}) = \mathbb{E}_\mathbb{Q}\left[ f\left( \frac{d\mathbb{P}}{d\mathbb{Q}} \right) \right] \geq \sup_{T \in \mathcal{T}} \{ \mathbb{E}_{s \sim \mathbb{P}}[T(s)] - \mathbb{E}_{s \sim \mathbb{Q}}[f^*(T(s))] \}$$

*where $\mathcal{T} \subseteq \{ T : \mathcal{S} \mapsto \mathbb{R} \}$ is any class of functions and $f^*(z) := \sup_{u \in \mathbb{R}}\{ u \cdot z - f(u) \}$ for any $z \in \mathbb{R}_+$ is the Fenchel conjugate.*

Utilizing Proposition 4, we propose the following min-max optimization problem to solve Objective (6).

**Theorem 5** ($f$-VI-IGL optimization problem). *Let $f_1$ and $f_2$ be functions satisfying the requirements in Proposition 4 and we denote by $f_1^*$ and $f_2^*$ their Fenchel conjugate, respectively. Objective (6) is equivalent to the following min-max optimization problem*

$$\min_{\psi \in \Psi} \max_{G \in \mathcal{G}} \min_{T \in \mathcal{T}} \Big\{ \mathbb{E}_{\mathbb{P}_{XAYR_\psi}}[G] - \mathbb{E}_{\mathbb{P}_{Y|R_\psi} \otimes \mathbb{P}_{XAR_\psi}}[f_1^*(G)]$$
$$- \beta \cdot \left( \mathbb{E}_{\mathbb{P}_{XAR_\psi}}[T] \right) - \mathbb{E}_{\mathbb{P}_{XA} \otimes \mathbb{P}_{R_\psi}}[f_2^*(T)] \Big) \Big\} \tag{7}$$

*where $G \in \mathcal{G} : \mathcal{X} \times \mathcal{A} \times \mathcal{Y} \times \{0, 1\} \mapsto \mathbb{R}$ and $T \in \mathcal{T} : \mathcal{X} \times \mathcal{A} \times \{0, 1\} \mapsto \mathbb{R}$.*

### 5.2 ALGORITHM DESCRIPTION

Here, we present $f$-VI-IGL Algorithm 1 as an optimization method to solve the $f$-VI-IGL optimization problem (7) for continuous random variables of the context $X$ and the feedback $Y$. The algorithm optimizes over three function classes $\mathcal{G}, \mathcal{T}$, and $\Psi$. Specifically, function class $\Psi = \{\psi_\theta\}_{\theta \in \Theta}$ consists of the reward decoders parameterized by $\theta \in \Theta$. Function class $\mathcal{G} = \{G_{\omega_1}\}$ parameterized by $\omega_1 \in \Omega_1$ is the estimator of $f_1$-MI $I_{f_1}(Y; X, A | R_{\psi_\theta})$. In addition, function class $\mathcal{T} = \{T_{\omega_2}\}_{\omega_2 \in \Omega_2}$ parameterized by $\omega_2 \in \Omega_2$ is the estimator of $f_2$-MI $I_{f_2}(X, A; R_{\psi_\theta})$. We focus on learning in the batch mode, where the algorithm has access to an offline dataset $\mathcal{D}_{\text{train}} = \{(x_t, a_t, y_t)\}_{t=1}^T$ consisting of the context-action-feedback tuples, which is collected by the behavior policy $\pi_b$ interacting with the environment.

At each epoch, $f$-VI-IGL first uses a mini-batch of data to estimate the value of Objective (7) (Lines 2-4). One difficulty is that estimating $I_{f_1}(Y; X, A | R_{\psi_\theta})$ requires sampling $(x, a, y) \sim \mathbb{P}_{R_{\psi_\theta}} \otimes \mathbb{P}_{Y|R_{\psi_\theta}} \otimes \mathbb{P}_{XA|R_{\psi_\theta}}$, where $\mathbb{P}_{Y|R_{\psi_\theta}}$ and $\mathbb{P}_{XA|R_{\psi_\theta}}$ can be intractable for continuous random variables of the context $X$ and the feedback $Y$. To address the problem, we first augment each data point $(x_t, a_t, y_t)$ for $N$ times to $\{(x_t, a_t, y_t, r_t^i)\}_{i=1}^N$, where $r_t^i \sim \text{Bernoulli}(\psi_\theta(x_t, a_t, y_t))$ and $N$ is a small positive integer (e.g. 5 in our experiments). To sample, e.g., the feedback $y \sim \mathbb{P}_{Y|R_{\psi_\theta}=1}$, we randomly sample a data point from $\{(x_t, a_t, y_t, r_t^j) : j \in [N], r_t^j = 1\}_{t=1}^T$, i.e., the "augmented" data points whose random decoded reward is 1. Given the estimated objective value, we alternatively update the parameters for the $f$-MI estimators and the reward decoder (Line 5). At the end of

---

**Algorithm 1** $f$-Variational Information-based IGL ($f$-VI-IGL)

---

**Require:** parameter $\beta > 0$, batch dataset $\mathcal{D}_{\text{train}} = \{(x_t, a_t, y_t)\}_{t=1}^T$ collected by the behavior policy $\pi_b$, reward decoders $\Psi = \{\psi_\theta : \mathcal{X} \times \mathcal{A} \times \mathcal{Y} \mapsto [0, 1]\}_{\theta \in \Theta}$, divergence measure $f_1$ for $I_{f_1}(Y; X, A | R_\psi)$ and the estimators $\mathcal{G} = \{G_{\omega_1} : \mathcal{X} \times \mathcal{A} \times \mathcal{Y} \times \{0, 1\} \mapsto \mathbb{R}\}_{\omega_1 \in \Omega_1}$, divergence measure $f_2$ for $I_f(X, A; R_\psi)$ and the estimators $\mathcal{T} = \{T_{\omega_2} : \mathcal{X} \times \mathcal{A} \times \{0, 1\} \mapsto \mathbb{R}\}_{\omega_2 \in \Omega_2}$.

1: **for** epoch $k = 1, 2, \cdots, K$ **do**
2:     Sample a mini-batch $\mathcal{D}_{\text{mini}} \sim \mathcal{D}_{\text{train}}$.
3:     Construct datasets with distributions $\mathbb{P}_Y \otimes \mathbb{P}_{R_{\psi_\theta}}$ and $\mathbb{P}_Y \otimes \mathbb{P}_{XAR_{\psi_\theta}}$ using $\mathcal{D}_{\text{mini}}$ (See the algorithm description).
4:     Estimate the $f$-MI terms

$$\widehat{I}_{f_1}(X, A; R_{\psi_\theta}) \leftarrow \mathbb{E}_{\mathbb{P}_{XAR_{\psi_\theta}}}[T] - \mathbb{E}_{\mathbb{P}_{XA} \otimes \mathbb{P}_{R_{\psi_\theta}}}[f_1^*(T)]$$

    and

$$\widehat{I}_{f_2}(Y; X, A | R_{\psi_\theta}) \leftarrow \mathbb{E}_{\mathbb{P}_{XAYR_{\psi_\theta}}}[G] - \mathbb{E}_{\mathbb{P}_{Y|R_{\psi_\theta}} \otimes \mathbb{P}_{XAR_{\psi_\theta}}}[f_2^*(G)]$$

5:     Alternatively update the parameters of $f$-MI estimators $T$ and $G$ (for a fixed $\psi_\theta$)

$$\omega_1 \leftarrow \omega_1 + \eta \cdot \nabla_{\omega_1}\left\{\widehat{I}_{f_1}(X, A; R_\psi)\right\}, \quad \omega_2 \leftarrow \omega_2 + \eta \cdot \nabla_{\omega_2}\left\{\widehat{I}_{f_2}(X, A; R_\psi)\right\}$$

    by gradient ascent and the reward decoder $\psi_\theta$ (for a fixed $T_{\omega_1}$ and $G_{\omega_2}$)

$$\theta \leftarrow \theta - \eta \cdot \nabla_\theta\left\{\widehat{I}_{f_1}(Y; X, A | R_{\psi_\theta}) - \beta \cdot \widehat{I}_{f_2}(X, A; R_{\psi_\theta})\right\}$$

    by gradient descent, where $\eta$ is the learning rate.
6: **end for**
7: Select between $\psi_\theta$ and its opposite counterpart $1 - \psi_\theta$ based on their decoded returns of $\pi_b$.
8: Train a policy $\pi$ via an offline contextual bandit oracle.
9: [**Output:** Policy $\pi$.]

---

the training, we use the learned reward decoder $\psi_\theta$ to train a policy via an offline contextual bandit oracle (Langford & Zhang, 2007; Dudik et al., 2011). However, note that in Objective (6), both the optimal reward decoder $\phi^*$ and its opposite counterpart $1 - \phi^*$ may attain the minimum simultaneously (while only one of them aligns is consistent with the true latent reward). Hence, we use the data-driven collector (Xie et al., 2021) and select the reward decoder (between the learned reward decoder $\psi_\theta$ and its opposite counterpart $1 - \psi_\theta$) that gives a decoded return of $\pi_b$ lower than 0.5.[1]

## 6 EMPIRICAL RESULTS

In this section, we empirically evaluate the $f$-VI-IGL algorithm on the number-guessing task (Xie et al., 2021) with noisy feedback, whose details are as follows.

**Number-guessing task with noisy feedback.** In the standard setting, a *random* image $x_t$ (context), whose corresponding number is denoted by $l_{x_t} \in \{0, 1, \cdots, 9\}$, is drawn from the MNIST dataset (Lecun et al., 1998) at the beginning of each round $t$. Upon observing $x_t$, the learner selects $a_t \in \{0, 1, \cdots, 9\}$ as the predicted number of $x_t$ (action). The latent binary reward $r_t = \mathbb{1}[x_t = l_{x_t}]$ is the correctness of the prediction label. Then, a *random* image of digit $r_t \in \{0, 1\}$ is revealed to the learner (feedback). In this paper, we consider four types of noisy feedback. Specifically, with probability $0.1$, the feedback is replaced with (i) *independent noises*: a random image of letter "t" (*True*) when the guess is correct or a random image of letter "f" (*False*) when the guess is wrong, which is sampled from the EMNIST Letter dataset (Cohen et al., 2017); (ii) *context-inclusive noises*: a random image of digit $(l_{x_t} + 6 \cdot r_t - 3) \bmod 10$, (iii) *action-inclusive noises*: a random image of digit $(a_t + 6 \cdot r_t - 3) \bmod 10$, (iv) *context-action-inclusive noises*: a random image of digit $(l_{x_t} + a_t + 6 \cdot r_t - 3) \bmod 10$. Note that the full conditional independence assumption does not

---

[1]Following the previous works (Xie et al., 2021; 2022), we assume the behavior policy has a low (true) return.

strictly hold as the feedback is also affected by the context-action pair (except for the independent noises).

**Data collection.** We focus on learning in the batch mode, where a training dataset $\mathcal{D}_{\text{train}} = \{(x_t, a_t, y_t)\}_{t=1}^T$ is collected by the uniform behavior policy using the *training set*. In all the experiments, the training dataset contains $60,000$ samples, i.e., $T = 60,000$. The output (linear) policy is evaluated on a test dataset $\mathcal{D}_{\text{test}}$ containing $10,000$ samples, which is collected by the uniform behavior policy from the *test set*. Additional experimental details are provided in Appendix A.

## 6.1 ROBUSTNESS TO NOISES

In this section, we show that $f$-VI-IGL is more robust to the noisy feedback than the previous IGL algorithm (Xie et al., 2021). We consider four types of noisy feedback in the number-guessing task. We report both the empirical prediction accuracy and the standard deviation in Table 1, where the VI-IGL method optimizes the standard MI-based Objective (4). We generate the same training and test datasets in these experiments and use the same initialization of the reward decoder and the linear policy. The results show that while the previous IGL works better when there is no noise (last row), VI-IGL attains more robust performance in all the noisy settings.

| Methods (ave±std) / Noises (0.1) | VI-IGL (Objective (4)) | IGL (Xie et al., 2021) |
|---|---|---|
| Independent | $\mathbf{63.4 \pm 18.1}$ | $25.0 \pm 18.4$ |
| Action-Inclusive | $\mathbf{62.8 \pm 21.2}$ | $21.6 \pm 12.4$ |
| Context-Inclusive | $\mathbf{69.2 \pm 13.3}$ | $15.4 \pm 14.3$ |
| Context-Action-Inclusive | $\mathbf{54.7 \pm 19.8}$ | $20.6 \pm 14.8$ |
| No Noises | $76.4 \pm 12.6$ | $\mathbf{82.2 \pm 4.3}$ |

Table 1: Robustness to Noises: The results are averaged over 16 trials.

**Why previous IGL method fails.** Recall that solving an appropriate reward decoder in the previous IGL method is given by (Xie et al., 2021, Assumption 2), which states that there exists a reward decoder that well distinguishes between the feedback (distribution) generated from a latent reward of 0 and the one generated from a latent reward of 1. When additional noises present in the feedback, these two distributions can be quite similar. For example, for context-inclusive noises, a latent reward of 0 can also generate an image of digit "1" ($l_{x_t} = 4$ and $r_t = 0$). Hence, the condition easily fails and the performance degrades.

## 6.2 ABLATION EXPERIMENTS

**1. Selection of $f$-divergences.** Recall that in Objective (6), we use $f_1$ and $f_2$ as general measures of $I(Y; X, A|R_\psi)$ and $I(X, A; R)$, respectively. We analyze how the selection of $f$-divergences affects the performance. We test three pairs of $f_1$-$f_2$: (i) KL-KL: both $f_1$ and $f_2$ are KL divergence, i.e., $f_1(x) = f_2(x) = x \log x$ (this case corresponds to Objective (4)), (ii) $\chi^2$-$\chi^2$: both $f_1$ and $f_2$ are Pearson-$\chi^2$ divergence, i.e., $f_1(x) = f_2(x) = (x-1)^2$, and (iii) $\chi^2$-KL: $f_1(x) = x \log x$ is KL divergence and $f_2(x) = (x-1)^2$ is Pearson-$\chi^2$ divergence. Note that in the last case, the objective value, i.e., $I_{\chi^2}(Y; X, A|R_\psi) - \beta \cdot I(X, A; R_\psi)$, upper bounds the value of Objective (4).[2] We summarize the results in Table 2 for a feedback-dependent reward decoder and $\beta = 1$. The results show that while different $f$-divergences benefit from different types of noises, both KL-KL and $\chi^2$-KL attain robust and consistent performance across these settings.

**2. Input of reward decoder.** We empirically analyze how the input of the reward decoder affects the actual performance. Particularly, we consider two types of input: (i) feedback $Y$ and (ii) context-action-feedback $(X, A, Y)$. We present the results in Table 3 for $\beta = \frac{1}{3}$ and $\chi^2$-KL divergence measure. The results show that while using a feedback-dependent reward decoder yields a higher expected value, a context-action-feedback-dependent reward decoder leads to a smaller standard deviation of the performance.

---

[2]By the inequality $\log \leq x-1$, we have that $D_{\text{KL}}(\mathbb{P}\|\mathbb{Q}) = \mathbb{E}_{\mathbb{P}}[\log(\frac{d\mathbb{P}}{d\mathbb{Q}})] \leq \mathbb{E}_{\mathbb{P}}[(\frac{d\mathbb{P}}{d\mathbb{Q}}-1)] = \mathbb{E}_{\mathbb{Q}}[(\frac{d\mathbb{P}}{d\mathbb{Q}})^2]-1 = D_\chi^2(\mathbb{P}\|\mathbb{Q})$.

| $f_1$-$f_2$ (ave±std;%) Noises (0.1) | KL-KL | $\chi^2$-$\chi^2$ | $\chi^2$-KL |
|---|---|---|---|
| Independent | $\mathbf{63.4 \pm 18.1}$ | $55.8 \pm 18.1$ | $\mathbf{63.5 \pm 14.3}$ |
| Action-Inclusive | $\mathbf{62.8 \pm 21.2}$ | $53.9 \pm 24.1$ | $58.5 \pm 23.4$ |
| Context-Inclusive | $\mathbf{69.2 \pm 13.3}$ | $57.3 \pm 25.7$ | $53.3 \pm 25.2$ |
| Context-Action-Inclusive | $54.7 \pm 19.8$ | $62.5 \pm 11.0$ | $\mathbf{66.4 \pm 17.9}$ |
| No Noises | $76.4 \pm 12.6$ | $\mathbf{77.2 \pm 9.0}$ | $71.7 \pm 11.5$ |

Table 2: Selection of $f$-divergences: The results are averaged over 16 trials.

| Input (ave±std;%) Noises (0.1) | $Y$ | $(X, A, Y)$ |
|---|---|---|
| Independent | $\mathbf{61.1 \pm 27.1}$ | $58.4 \pm 13.0$ |
| Action-Inclusive | $\mathbf{50.2 \pm 22.3}$ | $49.2 \pm 20.8$ |
| Context-Inclusive | $\mathbf{61.3 \pm 23.4}$ | $45.8 \pm 17.2$ |
| Context-Action-Inclusive | $\mathbf{59.8 \pm 26.0}$ | $55.2 \pm 20.4$ |
| No Noise | $\mathbf{68.0 \pm 19.4}$ | $54.3 \pm 20.0$ |

Table 3: Input of Reward Decoder: The results are averaged over 16 trials.

**3. Value of parameter $\beta$.** We present empirical results for different values of parameter $\beta$ in Objective (6). Recall that Objective (6) translates to $\arg\max_{\psi \in \Psi}\{I_{f_2}(X, A; R_\psi) - \beta^{-1} \cdot I_{f_1}(Y; X, A|R_\psi)\}$. Particularly, when $\beta = 0$, Objective (6) is equivalent to minimizing only $I_{f_1}(Y; X, A|R_\psi)$. Intuitively, a small $\beta$ may not work well for noisy feedback. We run experiments for $\beta = 0, \frac{1}{3}, 1, 2$ and provide the results in Table 4 for a feedback-dependent reward decoder and $\chi^2$-KL divergence measure. The results align with our intuition. Larger values of $\beta = 1, 2$ outperform smaller $\beta = 0, \frac{1}{3}$ in almost all cases, except for the context-inclusive noises. In addition, the regularization term $I_{f_2}(X, A; R_\psi)$ not only helps improve the robustness of the algorithm against noisy feedback but also benefits the training without noises, regarding both the empirical accuracy and the standard deviation. We find that $\beta = 1$ leads to consistently good performance.

| $\beta$ (ave±std;%) Noises (0.1) | $\beta = 0$ | $\beta = \frac{1}{3}$ | $\beta = 1$ | $\beta = 2$ |
|---|---|---|---|---|
| Independent | $63.7 \pm 24.8$ | $61.1 \pm 27.1$ | $63.5 \pm 14.3$ | $\mathbf{64.0 \pm 15.3}$ |
| Action-Inclusive | $50.1 \pm 28.2$ | $50.2 \pm 22.3$ | $\mathbf{58.5 \pm 23.4}$ | $55.3 \pm 23.7$ |
| Context-Inclusive | $52.0 \pm 28.4$ | $\mathbf{61.3 \pm 23.4}$ | $53.3 \pm 25.2$ | $56.1 \pm 20.2$ |
| Context-Action-Inclusive | $56.4 \pm 21.4$ | $59.8 \pm 26.0$ | $\mathbf{66.4 \pm 17.9}$ | $62.3 \pm 17.9$ |
| No Noises | $61.9 \pm 24.3$ | $68.0 \pm 19.4$ | $\mathbf{71.7 \pm 11.5}$ | $67.8 \pm 20.5$ |

Table 4: Value of Parameter $\beta$: The results are averaged over 16 trials.

## 7 DISCUSSION AND FUTURE WORK

We briefly discuss the limitations of our method. One limitation is the computation complexity. The $f$-VI-IGL algorithm needs to optimize over three function classes to find the desired reward decoder, which may consume a considerable computing budget. Hence, how to reduce the complexity remains to be explored. [An interesting future direction to our work is to develop a tight sample complexity bound for our proposed neural net-based reinforcement learning algorithm.] Another extension is to relax the full conditional independence assumption (Assumption 1). For example, Xie et al. (2022) consider the Action-Inclusive IGL (AI-IGL), where the feedback may also be affected by the action, i.e., $Y \perp\!\!\!\perp X|A, R$. In this case, the information-based objective can be $\arg\min_{\psi \in \Psi}\{I(Y; X|R_\psi, A) + \beta \cdot I(X, A; R_\psi)\}$.

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

## A  ADDITIONAL EXPERIMENTAL DETAILS

For the $f$-variational estimators (functions $T$ and $G$), the reward decoder $\psi$, and the linear policy $\pi$, we use a 2-layer fully-connected network to process each input image (i.e., the context or the feedback). Then, the concatenated inputs go through an additional linear layer and the final value is output. The same network structures are used to implement the reward decoder and the policy of the previous IGL algorithm (Xie et al., 2021). In each experiment, we train the $f$-VI-IGL algorithm for $1,000$ epochs with a batch size of $600$. Particularly, we alternatively update the parameters of the $f$-MI estimators and the reward decoders (i.e., $500$ epochs of training for each). For the previous IGL method, we follow the experimental details provided in the work of Xie et al. (Xie et al., 2021, Appendix C) and train the algorithm for 10 epochs over the entire training datasets.

## B  OVERFITTING WITHOUT REGULARIZATION

We show that when minimizing only $I(Y; X, A | R_\psi)$: **(A1)** *any* feedback-dependent reward decoder $\psi : \mathcal{Y} \to [0, 1]$ attains a small value, and **(A2)** the minimum is attained by a set of deterministic feedback-dependent reward decoders $\psi : \mathcal{Y} \to \{0, 1\}$ for environments where the feedback variable $Y$ is (nearly) deterministic to the context-action $(X, A)$. Both cases can lead to the "over-fitting" problem. To showcase these arguments, note that minimizing $I(Y; X, A | R_\psi)$ is equivalent to

$$\min_\psi \mathcal{L}'(\psi) := (H(R_\psi | X, A) - H(R_\psi | X, A, Y)) - (H(R_\psi) - H(R_\psi | Y))$$

where $H$ is the Shannon entropy. This holds by the facts that $I(Y; X, A | R_\psi) = I(Y; R_\psi | X, A) - I(Y; R_\psi) + I(Y; X, A)$ and the last term $I(Y; X, A)$ is *independent* of the reward decoder. Hence, the value is non-positive for any reward decoder $\psi : \mathcal{Y} \to [0, 1]$ by noting that $H(R_\psi | X, A, Y) = H(R_\psi | Y)$ and $H(R_\psi | X, A) \leq H(R_\psi)$. In contrast, $\mathcal{L}'$ can be positive for reward decoders that also depend on the context(-action), which justifies **(A1)**. Further, if the feedback $Y$ is (nearly) deterministic to the context-action $(X, A)$, we have that $H(R_\psi | X, A) \approx H(R_\psi | Y(X, A)) \approx 0$, and hence the minimum $-\log 2$ is attained by any reward decoder $\psi : \mathcal{Y} \to \{0, 1\}$ that assigns value 1 to "an arbitrary half of" the feedback in the data distribution (i.e., $\mathbb{E}_{x \sim d_0, a \sim \pi_b(\cdot|x)}[\psi(Y(x, a))] = 0.5$), which justifies **(A2)**.

