# OpenReview forum: "An Information Theoretic Approach to Interaction Grounded Learning"
_ICLR.cc/2024/Conference — Submitted to ICLR 2024_

### Official Review · Reviewer_yTyJ · 2023-11-03

**Soundness:** 3 good
**Presentation:** 2 fair
**Contribution:** 3 good
**Rating:** 6
**Confidence:** 3

**Summary:**

This paper introduces a novel algorithm for the Interaction Grounded
Learning (IGL) problem based on an information theoretic
objective. The algorithm is shown to demonstrate superior performance
under several noise-corrupted adaptations of a standard benchmark
relative to the state of the art.

**Strengths:**

This paper presents a well-motivated algorithm for a relatively new
problem in ML. This algorithm addresses the challenge of dealing with
significant randomness in the feedback given to the learner, and the
empirical results confirm the benefits of the proposed
approach. Ablations are also given to provide more insight about which
parts of the algorithm had notable effects on performance.

**Weaknesses:**

The main weaknesses that I can imagine are with respect to the
organization of the paper. Perhaps I am biased, since I had not
previously been aware of the IGL framework, but I had trouble
understanding both the motivation behind the problem framework and the
algorithm. Given that IGL is a relatively new and unexplored topic, I
believe the paper can be improved by giving less abstract examples of
the IGL framework early on (for instance, everything became quite a
bit more clear to me once I saw the benchmark in the experimental
section).

Moreover, I did not find the argument for the necessity to include the
regularization term particularly convincing. I would have liked to
have seen stronger evidence (particularly before the regulization term
was introduced) to suggest that the mode of overfitting discussed in
the paper actually occurs. Notably, without the regularization term,
the algorithm/optimization problem is considerably simpler.
Having said that, since the algorithm still
appears to be novel even without the regularization term, I think this
is mostly an issue of organizing the content to improve clarity.

**Questions:**

Does the context distribution have to be fixed throughout training (or
can it, for example, be adversarial)?

I believe there is a mistake in the notation of $V(\pi)$, particularly
with $(x, a)\sim d_0\otimes\pi$. This looks like $x, a$ are sampled
independently, but really $a$ sampled conditionally on $x$. I believe
it should be more like $x\sim d_0,a\sim\pi(\cdot\mid x)$.

Admittedly I am not intimately familiar with MI optimization, but it
is not obvious to me why we should expect minimization of the standard
MI objective to "overfit" to maximize $I(Y; R_\psi)$ as you claim. I
see that the objective can be decreased by increasing this term, but I
see now reason why thi wouldn't be balanced by a decrease of the $I(Y;
X, A, R_\psi)$
term. Has this been demonstrated experimentally? If so, can you
provide citations? If this paper is the first demonstration of this
overfitting phenomenon, it might be nice to show those results
before defining the regularized objective, it would help motivate your approach.

Why is it that the experiments with $\beta=0$ exhibited the most
variance? In this case, I would expect variance to be lower, since
you're optimizing over fewer neural network parameters / the
optimization has one less level of nesting.

In Algorithm 1, what does "Ensure: Policy $\pi$" mean?

---

> ### Author Response · Authors · 2023-11-18
> **Response to Reviewer yTyJ**
>
> We thank Reviewer yTyJ for his/her time and feedback on our work. The following is our response to the reviewer's comments and questions.
>
> **1. Regularizer in Objective (4) and "over-fitting" problem.**
>
> The reviewer asks for stronger evidence for including the regularizer $I(X,A;R_\psi)$ in Objective (4) and explanations for the "over-fitting" problem. As this has been empirically shown in our numerical results in Table 4, we would like to discuss it in greater details. Indeed, we can show that when minimizing only $I(Y;X,A|R_\psi)$: **(A1)** *any* feedback-dependent reward decoder $\psi:\mathcal{Y}\to[0,1]$ attains a small value, and **(A2)** the minimum is attained by a subset of feedback-dependent reward decoders $\psi:\mathcal{Y}\to\{0,1\}$ that assigns value 1 to "an arbitrary half of" the feedback in the data distribution, if the feedback variable $Y$ is (nearly) deterministic to the context-action $(X,A)$. Both cases would lead to the "over-fitting" problem. We totally agree with the reviewer that presenting these results before defining the regularized objective helps motivate our approach. We have included this discussion in the revised Section 4.1.
>
> **Detailed analysis.** To showcase these arguments, first note that minimizing $I(Y;X,A|R_\psi)$ is equivalent to
> $$\min_\psi\mathcal{L}'(\psi):=(H(R_\psi|X,A)-H(R_\psi|X,A,Y))-(H(R_\psi)-H(R_\psi|Y))$$
> where $H$ is the Shannon entropy. This holds by the facts that $I(Y;X,A|R_\psi)=I(Y;R_\psi|X,A)-I(Y;R_\psi)+I(Y;X,A)$ and the last term $I(Y;X,A)$ is independent of the reward decoder. Hence, the value is non-positive for any reward decoder $\psi:\mathcal{Y}\to[0,1]$ by noting that $H(R_\psi|X,A,Y)=H(R_\psi|Y)$ and $H(R_\psi|X,A)\le H(R_\psi)$. In contrast, $\mathcal{L}'$ can be positive for reward decoders that also depend on the context(-action), which justifies **(A1)**. Further, if the feedback $Y$ is (nearly) deterministic to the context-action $(X,A)$, we have that $H(R_\psi|X,A)\approx H(R_\psi|Y(X,A))\approx0$, and hence the minimum $-\log2$ is attained by any reward decoder $\psi:\mathcal{Y}\to\{0,1\}$ that assigns value 1 to "an arbitrary half of" the feedback in the data distribution (i.e., $\mathbb{E}\_{x\sim d_0,a\sim\pi_\textup{b}(\cdot|x)}[\psi(Y(x,a))]=0.5$), which justifies **(A2)**.
>
> **2. Large variance for experiments with $\beta=0$.**
>
> We believe that the observation that the experiments with $\beta=0$ displayed the highest variance is closely related to the minimization of $I(Y;X,A|R_\psi)$, which we have discussed in response to the reviewer's Comment 1 above. Since any feedback-dependent reward decoder can attain a small value of $I(Y;X,A|R_\psi)$, the training process will be highly unstable and the optimized reward decoder will be constantly changing at every iteration that could lead to high variance.
>
> **3. "Does the context distribution have to be fixed throughout training?"**
>
> Yes, while the results are reported for fixed context distribution throughout training, the algorithm can be also applied to non-stationary context distribution.
>
> **4. Notation of $V(\pi)$.**
>
> We thank the reviewer for pointing out the possible ambiguity in the notation. We have revised it to $V(\pi):=\mathbb{E}\_{x\sim d_0}\mathbb{E}_{a\sim\pi(\cdot|x)}[\mu(x,a)]$ in the paper.
>
> **5. "Ensure: Policy $\pi$" in Algorithm 1.**
>
> We thank the reviewer for pointing out the typo in Algorithm 1. We have corrected this typo in the revision.

---

### Official Review · Reviewer_Jvuq · 2023-11-05

**Soundness:** 2 fair
**Presentation:** 2 fair
**Contribution:** 1 poor
**Rating:** 3
**Confidence:** 5

**Summary:**

In this paper, the authors proposed to leverage information-theoretical quantities to solve the interaction grounded learning in the noisy scenarios. The main contribution of this paper is a new objective based on the mutual information. However, how to estimate the mutual information has been heavily studied in the literature, and I don’t see any new components here. Finally, the authors do not provide the appendix.

**Strengths:**

N/A.

**Weaknesses:**

* Although the intuition of such objective is sound, I don’t think there are any rigorous theoretical justification, e.g. the statistical error on estimating the reward with different level of noise.
* I believe the authors ignore large amounts of work on (conditional) mutual information estimation, that covers most of the theoretical derivation in the paper, e.g. [1][2].

[1] Song, Jiaming, and Stefano Ermon. "Understanding the limitations of variational mutual information estimators." arXiv preprint arXiv:1910.06222 (2019).
[2] Poole, Ben, et al. "On variational bounds of mutual information." International Conference on Machine Learning. PMLR, 2019.

* The authors do not provide the appendix, which should be a crucial issue that can lead to the rejection.

**Questions:**

To echo the weakness, I would like to ask:
* Is there any rigorous theoretical guarantee for motivating this objective?
* Can the authors discuss the relationship between the proposed estimation method and the existing work, not limited to the references I provide?

---

> ### Author Response · Authors · 2023-11-18
> **Response to Reviewer Jvuq**
>
> We thank Reviewer Jvuq for his/her feedback on our work. The following is our response to the reviewer's comments.
>
> **1. Finite-Sample Guarantees.**
>
> Please refer to our general response.
>
> **2. Novelty compared to the existing work.**
>
> We note that the papers mentioned by the reviewer focus on the variational estimation of mutual information. On the other hand, our work focuses on the specific RL problem in the IGL setting. Please note that our main contribution is to propose an information theoretic approach to this RL problem, while the reviewer seems to evaluate the novelty of our estimation method for conditional mutual information. We sincerely hope that the reviewer would also consider our contributions on addressing the IGL-based RL setting in his/her evaluation of our work.
>
> **3. Appendix**
>
> We wish the reviewer could be more specific on what is missing in the main text. Since all our theoretical derivations and numerical results have been already included in the main text, we did not include an Appendix in the submission. According to the conference rules, submitting an Appendix is not mandatory for the submissions. We will be happy to answer any follow-up questions the reviewer might have on what he/she thinks is missing in the paper. Also, we have provided our code in the revision as a supplementary file, which can be used to verify the numerical results if necessary.

---

### Official Review · Reviewer_8FsD · 2023-11-06

**Soundness:** 2 fair
**Presentation:** 2 fair
**Contribution:** 2 fair
**Rating:** 5
**Confidence:** 4

**Summary:**

this paper proposed an information-theoretical method for enforcing conditional independence between Context and A, given latent reward variable. The method is generalize to f -Information measures.

**Strengths:**

an interesting problem is considered. the Preliminaries section provided a good background.

**Weaknesses:**

many steps in the derivation / explanation are not explained. For example, in my option the quantity (without an equation number) after Eq (4) does not follow from Eq 4. Is Theorem 3 trivial that it does not require proof?

**Questions:**

Why is it important to show results for a few different f-divergences?
Why were these f-divergences chosen?
How to chose "f-divergences"? Table 2 show that different scenarios requires different "f-divergences".

---

> ### Author Response · Authors · 2023-11-18
> **Response to Reviewer 8FsD**
>
> We thank Reviewer 8FsD for his/her time and feedback on our work. The following is our response to the reviewer's comments and questions.
>
> **1. Quantity after Equation (4).**
>
> We thank the reviewer for pointing out the typo in the quantity after Equation (4). In the revision, we corrected the typo and replace $I(X,A|R_\psi)$ with $I(X,A;R_\psi)$.
>
> **2. Derivation of Theorem 3.**
>
> We note that the derivation of Theorem 3 has been already explained in Section 4.2. Based on the reviewer's question, we have further discussed the derivation of Theorem 3 in Section 4.2 of the revised text.
>
> **3. Selection of $f$-divergences.**
>
> Different $f$-divergences have been shown to lead to different convergence and generalization behaviors [1, 2]. As a result, a proper choice of $f$-divergence could heavily depend on the target RL setting. This point is confirmed by the results in Table 2 showing that the optimal performance in different RL problems can ne attained by the algorithms based on different $f$-divergences.
>
> [1] Nowozin, Sebastian, Botond Cseke, and Ryota Tomioka. "f-gan: Training generative neural samplers using variational divergence minimization." Advances in neural information processing systems 29 (2016).
>
> [2] Rubenstein, Paul, et al. "Practical and consistent estimation of f-divergences." Advances in Neural Information Processing Systems 32 (2019).

---

### Official Review · Reviewer_kBCY · 2023-11-08

**Soundness:** 3 good
**Presentation:** 3 good
**Contribution:** 3 good
**Rating:** 5
**Confidence:** 2

**Summary:**

The paper discusses the challenges and solutions in the context of reinforcement learning (RL) algorithms when the agent lacks complete knowledge of the reward variable. When there is no explicit reward, the agent must infer the reward from observed feedback, increasing the computational and statistical complexity of the RL problem.

To address these challenges, the Interaction-Grounded Learning (IGL) framework is introduced. In IGL, the agent observes a context, takes an action, and receives feedback, aiming to maximize the unobserved return by inferring rewards from this interaction. The key to this approach is a properly inferred reward decoder, which maps the context-action-feedback tuple to a prediction of the latent reward. However, learning such a reward decoder can be information-theoretically infeasible without additional assumptions on the relationship between context, action, feedback, and reward variables.

One such assumption, known as Full Conditional Independence, posits that feedback is conditionally independent of context and action given the latent reward. Existing IGL methods use this assumption and propose joint training of the policy and decoder. However, noisy feedback in real-world scenarios may challenge the validity of this assumption.

The paper introduces Variational Information-based IGL (VI-IGL) as an information-theoretic approach to IGL-based RL tasks. VI-IGL aims to ensure conditional independence between feedback and context action by minimizing an information-based objective function. It includes a regularization term to make the reward decoder robust to feedback noise.

The challenge of optimizing this objective is addressed by leveraging the variational representation of mutual information (MI) and formulating the problem as a min-max optimization. This allows gradient-based algorithms to efficiently solve it. The paper also extends the approach to f-Variational Information-based IGL (f-VI-IGL), creating a family of algorithms for the IGL-based RL problem.

Empirical results suggest that VI-IGL outperforms existing IGL RL algorithms, particularly in noisy feedback scenarios. The key contributions of the paper include the introduction of an information-theoretic approach to IGL-based RL, a novel optimization technique for handling continuous random variables, and the extension of the approach to f-VI-IGL.

**Strengths:**

- The paper proposes a novel method to solve a real problem when we need to apply RL algorithms in real applications.

- The paper provides a clear context of what is already done in previous literature and what are the main challenges.

- The authors explain appropriately the novel method, providing the preliminary to understand the proposed approach. The novel algorithm uses a regularized information-based IGL objective. Then they provide a more tractable objective using the variational information-based approach.

- The authors compare the proposed algorithm with SotA method, showing that the proposed approach outperforms previous ones when there is noise.

**Weaknesses:**

- The paper does not theoretically discuss how the changes in the (4) objective can lead to worse performances. How much the KL approximation optimum can be far from the original optimal solution?

- The paper is lacking theoretical results (e.g. sample complexity or regret analysis). Xie et al. provide a sample complexity result for their proposed algorithm. Could you derive similar results?

- Experimental evaluation:

    - Why does the proposed method achieve worse results in the No noises setting compared to Xie et Al.?

- In the conclusion the authors mentioned that the algorithm is computationally expensive. Could you compare the computational complexity of the proposed method with the one of Xie et Al.?

**Questions:**

See weaknesses.

---

> ### Author Response · Authors · 2023-11-18
> **Response to Reviewer kBCY**
>
> We thank Reviewer kBCY for his/her time and feedback on our work. Here is our response to the reviewers' comments and questions:
>
> **1. "How much the KL approximation optimum can be far from the original optimal solution?"**
>
> We follow a standard approach to apply variational analysis to estimating the  mutual information. In Proposition 2, if the function class $\mathcal{T}$ includes all functions, then the KL approximation is tight. As our proposed algorithm leverages neural networks and gradient-based method, we adopt the  parameterized function class. We note that lower bounding the mutual information by the variational optimization is standard [1, 2]. For a comprehensive review on this topic, please refer to the work of [3].
>
> [1] Nowozin, Sebastian, Botond Cseke, and Ryota Tomioka. "f-gan: Training generative neural samplers using variational divergence minimization." Advances in neural information processing systems 29 (2016).
>
> [2] Belghazi, Mohamed Ishmael, et al. "Mutual information neural estimation." International conference on machine learning. PMLR, 2018.
>
> [3] Nguyen, XuanLong, Martin J. Wainwright, and Michael I. Jordan. "Estimating divergence functionals and the likelihood ratio by convex risk minimization." IEEE Transactions on Information Theory 56.11 (2010): 5847-5861.
>
> **2. Finite-Sample Guarantees.**
>
> Please refer to our general response.
>
> **3. Numerical results in the noiseless setting.**
>
> We note that the algorithm proposed by Xie et al. (2021) is designed for the noiseless IGL setting, while our proposed algorithm $f$-VI-IGL is designed to address the RL task under a noisy feedback, which is often the case in real-world scenarios. As we observed in our numerical results, while Xie et al.'s algorithm attains better numerical results in the noiseless setting, its performance can degrade significantly under a noisy observation of the feedback variable. In contrast, our proposed method managed to achieve better results in the noisy case.
>
> **4. Computational costs of the proposed RL method.**
>
> While our proposed $f$-VI-IGL algorithm requires training of deep neural nets to estimate the mutual information terms in the objective, we observed that the training often enjoys a fast convergence. In our experiments, the training times of $f$-VI-IGL were in worst case $2\times$ more than the training time of Xie et al.'s algorithm, while achieving satisfactory numerical results in the noisy IGL environment.

---

### Official Review · Reviewer_1D19 · 2023-11-10

**Soundness:** 3 good
**Presentation:** 3 good
**Contribution:** 3 good
**Rating:** 6
**Confidence:** 2

**Summary:**

This paper designs an algorithm for inverse RL using the properties of f-divergence. It conducts experiment to validate the algorithm.

**Strengths:**

1. The idea of using f-divergence is novel.

2. The results of the experiment validates their algorithm.

**Weaknesses:**

1. The proposed algorithm does not have finite-sample theoretical guarantee.

**Questions:**

See the 'weakness' section.

---

> ### Author Response · Authors · 2023-11-18
> **Response to Reviewer 1D19**
>
> We thank Reviewer 1D19 for his/her time and feedback on our work. The following is our response to the reviewer's comments and questions.
>
> **1. Finite-sample guarantees**
>
> Please refer to our general response.

---

### Author Response · Authors · 2023-11-18
**General Response**

We would like to thank the reviewers for their time and constructive feedback. The following is our general response to the reviewer's comments on the finite sample complexity guarantee. We have responded to the reviewers' other questions and comments in the response to the individual reviews.

**Finite sample complexity guarantees**

The main focus of our work is to derive and propose a mutual-information-based algorithm to address the IGL-based RL problem. As discussed in the paper, our proposed algorithm leverages deep neural networks for the estimation of the mutual information terms from training data. Similar to standard applications of deep neural nets to supervised learning tasks, we observed satisfactory numerical performance of our neural net-based RL algorithm despite the large capacity of the neural net to memorize training data. As already discussed in the literature [1,2], the application of a uniform convergence bound to such overparameterized function spaces, e.g. the bounds in terms of covering number and complexity measures, will lead to a vacuous generalization bound with no nontrivial implications on the performance of the learning algorithm.

Therefore, the derivation of finite-time sample complexity guarantees requires *tight generalization bounds* for overparameterized neural networks, which is still an open problem in the learning theory literature and beyond the scope of our theoretical study. However, we agree with the reviewers on the importance of sample complexity guarantees for neural net-based RL methods, which could be an interesting future direction to our work. We have added this discussion to the revised conclusion section.

[1] Zhang et al, "Understanding deep learning requires rethinking generalization", ICLR 2017.

[2] Nagarajan and Kolter, "Uniform convergence may be unable to explain generalization in deep learning", NeurIPS 2021.

---

### Author Response · Authors · 2023-11-23

Dear Reviewers,

We appreciate and thank you for the time and effort in reviewing our work. As the discussion period will end today, we would like to kindly ask if there are any remaining questions or comments regarding our responses or, in general, our revised draft. We will be happy to discuss and clarify if any detail remains unclear to the reviewers.

---

### Meta-Review · Area_Chair_ydrY · 2023-12-05

**Metareview:**

The reviewers are in agreement that the manuscript is limited by a lack of discussion in how changes in the learning objective can lead to worse performances, what is the role of KL divergence approximation, and a theoretical analysis of the proposed approach in general. In addition, concerns regarding performance of the algorithm in the absence of noise and a lack of through discussion of algorithm complexity prevent it from being publishable at this time.

**Justification For Why Not Higher Score:**

NA

**Justification For Why Not Lower Score:**

NA

---

### Decision · Program_Chairs · 2024-01-16

Reject